# The effect of sodium-glucose cotransporter 2 inhibitors in patients with chronic kidney disease with or without type 2 diabetes mellitus on cardiovascular and renal outcomes: A systematic review and meta-analysis

Carlos Ignacio Reyes-Farias[1]o, Marcelo Reategui-Diaz[1]o *, Franco Romani-Romani[1], Larry Prokop[2]

1 Faculty of Medicine, Universidad de Piura, Lima, Lima, Peru, 2 Knowledge and Evaluation Research Unit, Mayo Clinic, Rochester, Minnesota, United States of America

o These authors contributed equally to this work.

* fabio.reategui@alum.udep.edu.pe

## Abstract

### Background

Sodium-glucose cotransporter 2 (SGLT-2) inhibitors have shown a favorable effect on cardiovascular and renal outcomes in patients with type 2 diabetes mellitus (T2DM). However, their efficacy in patients with chronic kidney disease (CKD) with or without T2DM has not yet been analyzed.

### Objective

To assess the cardiovascular and renal effects of SGLT-2 inhibitors in patients with CKD with and without T2DM, including all CKD patients in the current literature.

### Methods

We searched MEDLINE, EMBASE, CENTRAL and Scopus for randomized controlled trials of SGLT-2 inhibitors that evaluated cardiovascular and kidney outcomes in patients with CKD, or trials in which these patients were a subgroup. We defined 2 primary outcomes: a composite of cardiovascular death or hospitalization for heart failure, and a composite renal outcome. For each outcome, we obtained overall hazard ratios with 95% confidence intervals by using a random effects model.

### Results

We included 14 randomized controlled trials. SGLT-2 inhibitors decreased the hazard for the primary cardiovascular outcome (HR 0.76; [95% CI 0.72–0.79]) and the primary renal outcome (HR 0.69; [95% CI 0.61–0.79]) in patients with CKD with or without T2DM. We did

**Data Availability Statement:** All relevant data are within the paper and its Supporting Information files.

**Funding:** The author(s) received no specific funding for this work.

**Competing interests:** The authors have declared that no competing interests exist.

not find significant differences in the subgroup analyses according to diabetes status, baseline eGFR values or the type of SGLT-2 inhibitor used.

## Conclusion

In patients with CKD, treatment with SGLT-2 inhibitors in addition to standard therapy conferred protection against cardiovascular and renal outcomes. Further research on patients with non-diabetic CKD should be done to confirm the utility of these medications in this population. (PROSPERO ID: CRD42021275012).

## Introduction

Chronic kidney disease (CKD) is an important cause of mortality worldwide [1]. Until recently, only angiotensin-converting enzyme (ACE) inhibitors and angiotensin-receptor blockers (ARBs) had been shown to decrease proteinuria and prevent progression to end-stage kidney disease [2,3].

Sodium-glucose cotransporter 2 (SGLT-2) inhibitors are used to treat people with type 2 diabetes mellitus (T2DM) [4,5]. They control glycemia and HbA1c by inhibiting the SGLT-2 cotransporter in proximal tubules, responsible for the reabsorption of the majority of the filtered glucose in the glomeruli. However, SGLT-2 inhibitors have other non-glycemic effects, such as lowering glomerular hyperfiltration and proteinuria, and anti-inflammatory properties in the glomeruli [6–9]. The relevance of these mechanisms has been reflected in the DAPA-CKD [10] and EMPA-KIDNEY [11] trials, which enrolled both diabetic and non-diabetic patients with CKD. These trials have shown a favorable effect of the use of SGLT-2 inhibitors on some cardiovascular and renal outcomes, whether the kidney injury was associated with T2DM or not. However, the evidence in patients with CKD without T2DM is still scarce.

Previous systematic reviews have demonstrated the efficacy of SGLT-2 inhibitors in reducing the risk of cardiovascular and renal events in patients with CKD and T2DM [12–15]. One systematic review included studies in both diabetic and non-diabetic CKD patients [16]. However, to date, there has been no comprehensive systematic review assessing the impact of SGLT-2 inhibitors on the entire CKD population, as provided in the available literature. Incorporating data from all CKD patients within clinical trials, not solely those designed exclusively for CKD patients but also those in which CKD patients were included as a subgroup, could enhance the precision of the effect estimates and improve the applicability of the findings.

The aim of this systematic review is to assess the cardiovascular and renal effects of SGLT-2 inhibitors in all available CKD patients in the current literature and to evaluate whether the presence of T2DM modifies their overall impact. This may help clarify a new therapeutic role for these drugs.

## Methods

To conduct this study, we followed the Preferred Reporting Items for Systematic Reviews and Meta-analyses (PRISMA) 2020 guideline [17]. The protocol of this systematic review and meta-analysis has been registered in PROSPERO (CRD42021275012).

### Eligibility criteria

We included randomized controlled trials of SGLT-2 inhibitors that evaluated cardiovascular and kidney outcomes in patients with CKD, as well as trials in which CKD patients were

included as a subgroup. For the latter, only the information from patients with CKD was extracted and used in this study. There were no restrictions regarding the type of SGLT-2 inhibitor, and the comparator group could receive either placebo and standard therapy or standard therapy alone. Patients needed to be adults with CKD, regardless of the definition of CKD, and with or without T2DM. Studies were included regardless of differences in time to follow-up.

We defined two primary outcomes. First, the cardiovascular composite outcome included hospitalization for heart failure (HF) or cardiovascular (CV) death. Second, a renal composite outcome was defined as doubling of serum creatinine, decline of eGFR > 50%, end-stage kidney disease, renal replacement therapy, transplantation, or renal death. The different definitions of primary outcomes are presented in the Supplement (S1 Table). The secondary outcomes were all-cause mortality; major adverse cardiovascular events (MACE, which consists of non-fatal stroke, non-fatal myocardial infarction and CV death); hospitalization for HF; and CV death.

## Information sources and search strategy

A comprehensive search from each database's inception to February 24th, 2023, was conducted with no language restriction. The databases included Ovid MEDLINE(R) and Epub Ahead of Print, In-Process & Other Non-Indexed Citations, and Daily, Ovid EMBASE, Ovid Cochrane Central Register of Controlled Trials (CENTRAL), and Scopus. The search strategy is detailed in the Supplement (S2 Table). We searched for additional records in the references section of the selected studies.

## Screening of studies

Before the selection process, we performed a pilot test of the eligibility criteria to reach consistency between reviewers. The abstract screening and the full-text assessment for eligibility was done independently by two authors (C.R., M.R.) with DistillerSR Version 2.40.0 (Evidence Partners; 2022). The final decision was confirmed by a third person (O.J.P.) in case of disagreement.

## Data extraction

**Data collection and author contact.**   Data extraction was performed independently and individually by C.R. and M.R. using Microsoft® Excel® Version 2208. Then, both authors crosschecked each other's extracted information to ensure consistency; if a disagreement was found, both reviewed the full-text article again. We requested unreported outcome data from the corresponding authors of each study and included the information they provided us in the analyses. We extracted the corresponding authors' information and contacted them via e-mail once. Out of 15 emails sent, seven authors responded. Three authors shared published research papers which we had not retrieved with our search strategy.

**Data items.**   Two authors independently extracted the following data of each study: publication year, trial registration number, inclusion and exclusion criteria, type of SGLT-2 inhibitor, comparator type (placebo or standard care) and follow-up time (years), patient baseline characteristics and clinical outcomes (primary and secondary outcomes as described above), and risk of bias indicators (shown in the section below). The baseline characteristics were sex, age (years), race, diabetes status (T2DM, no T2DM), diabetes medications (insulin, metformin, sulfonylurea, DPP-4 inhibitor, GLP-1 receptor agonist), HbA1C (percentage), CV history (any CV disease and HF) and CV medications (ACE inhibitors or ARB, β-Blocker, statin and diuretic), and renal baseline characteristics (estimated glomerular filtration rate [eGFR] < 60

ml/min/1.73m2 and urine albumin-to-creatinine ratio [UACR] $\geq$ 300 mg/g). We only considered the data from the patients with CKD according to the definition of each study and/or patients with an eGFR $<$ 60 ml/min/1.73 m$^2$ (S1 Table).

**Risk of bias assessment.** Study quality was assessed independently and individually by C. R. and M.R. using the Cochrane Risk of Bias Tool 2. This tool consists of five domains (randomization process, deviations from the intended interventions, missing outcome data, measurement of the outcome, and selection of the reported result) [18].

We used the two primary outcomes for our evaluation and rated the overall risk of bias of each study as 'low', 'some concerns', or 'high'. Any disagreement was discussed between the authors to reach a consensus. We generated summary charts with the web-based version of the *robvis* package [19].

## Quality of the evidence

Two authors (C.R., M.R.) independently assessed the quality of the evidence using the Grading of Recommendations Assessment, Development, and Evaluation (GRADE) system [20]. Disagreements were resolved by discussion.

The GRADE system consists of five domains (risk of bias, imprecision, inconsistency, indirectness, and publication bias) to assess the level of certainty of the evidence for each outcome as high, moderate, low, or very low. We considered a large magnitude of effect (Hazard ratio [HR] $<$ 0.5), a dose-response gradient and a plausible confounding as criteria to increase the level of certainty [21].

We used the GRADEpro GDT software to elaborate 'Summary of the evidence' table [22]. All the reasons to rate up or down the quality of the evidence were documented as footnotes.

## Statistical analysis

Patient baseline characteristics were summarized according to the type of variable. For qualitative variables, the frequency and percentage were reported; for quantitative variables, the mean and the standard deviation were used instead. If data was not directly available for the subgroup of all CKD patients in a study, we calculated it from the separate intervention and control groups. For the qualitative variables, we added up the frequencies and calculated the percentages based on the total number of participants. For the quantitative variables, we calculated the weighted means and weighted standard deviations, considering the intervention and control groups sizes.

Time-to-event CV and renal outcomes are presented as incidence rate (events per 100 person-years or events per 1000 person-years), and when appropriate, as cumulative incidence (events/total participants). All outcomes were summarized by using the reported HR and their corresponding 95% confidence intervals (CI) in each study.

We performed a meta-analysis including all studies with available data. We fit the random effects model with the Paule-Mandel estimation of the tau-squared [23], since it was more appropriate for binary outcomes than the restricted maximum likelihood method we originally considered. The heterogeneity was adjusted with the Hartung-Knapp method which accounts for the uncertainty in the estimate of tau-squared [24]. The I-squared statistic was calculated to assess the percentage of total variability attributed to heterogeneity among the true effects in our meta-analysis [25].

When a study reported effect estimates (HR) for complementary subgroups from within an outcome such as varying levels of eGFR (45 to $<$ 60 ml/min/1.73 m$^2$ and eGFR $<$ 45 ml/min/1.73 m$^2$), we aggregated these subgroups with a fixed effects model as described by Borenstein

*et al.* [26]. For instance, based on our example a new HR was generated for patients with eGFR < 60ml/min/1.73 m$^2$.

We performed subgroup analyses to detect differences in the effect of SGLT-2 inhibitors in patients according to the presence of T2DM, baseline eGFR values, and the type of SGLT-2 inhibitor. We assessed the interaction between the subgroups with a Wald-type test [27].

For each analysis we used the most comprehensive definition of CKD available in each study, which could include patients with either a reduced eGFR or significant albuminuria (S2 Table). If this definition was not provided, we established an eGFR < 60 ml/min/1.73 m$^2$ as diagnostic of CKD and included these patients in the meta-analysis.

Although this analysis was not described in our protocol, if the Hartung-Knapp adjustment was suspected to be interfering with the precision of a confidence interval, we performed a sensitivity analysis without the adjustment to confirm the result. Additionally, we performed a sensitivity analysis to re-assess our main meta-analysis only in patients with CKD defined as eGFR < 60 ml/min/1.73 m$^2$.

Publication bias was explored using contour enhanced funnel plots for the primary and secondary outcomes. Additionally, we performed the Egger's regression test to statistically assess the risk of publication bias [28]. We originally intended to explore publication bias in outcomes with 10 or more studies; however, the large sample size of the included trials allowed us to evaluate all outcomes.

All the analyses were done with R version 4.2.1 [29] and RStudio version 2022.07.1+554 [30] using the *meta* package [31].

## Results

### Study selection

We retrieved 5009 records from databases. After duplicate removal (n = 1534) and abstract screening (n = 3475) we assessed 250 full-text articles for eligibility. Of these, 24 articles that corresponded to 13 clinical trials were included [10,11,32–53].

We searched for other documents by reviewing citations from included articles. With this strategy we identified four articles with additional outcome data from included trials [54–57], as well as two articles from a trial not previously found [58,59]. Furthermore, we obtained five articles from included trials through author contact [13,15,60–62]. In total, we included 35 articles from 14 trials (Fig 1).

### Baseline characteristics

We included 44 593 patients from 14 randomized placebo-controlled clinical trials [10,11,32–40,52,56,58] of five SGLT-2 inhibitors (empagliflozin [n = 4], canagliflozin [n = 3], dapagliflozin [n = 4], ertugliflozin [n = 1], and sotagliflozin [n = 2]) in patients with CKD. S3 Table shows the characteristics of the patients.

Out of the 14 studies included in the quantitative synthesis, five included only patients with CKD, and nine also included non-CKD patients as well. For the latter we only extracted the data from the CKD subgroups. Furthermore, six studies included patients with and without T2DM, while eight only considered patients with T2DM. Finally, only two studies made comparisons between patients with CKD and T2DM and patients with CKD without T2DM. For the subgroup analysis, except for the primary renal outcome, we only used data from the DAPA-CKD trial for the non-diabetic subgroups, since there was no other study that reported this data. Of note, every study except for two (the EMPEROR-Reduced and EMPEROR-Preserved trials) excluded patients with T1DM. The median follow-up ranged from 0.8 to 4.2 years across all the trials.

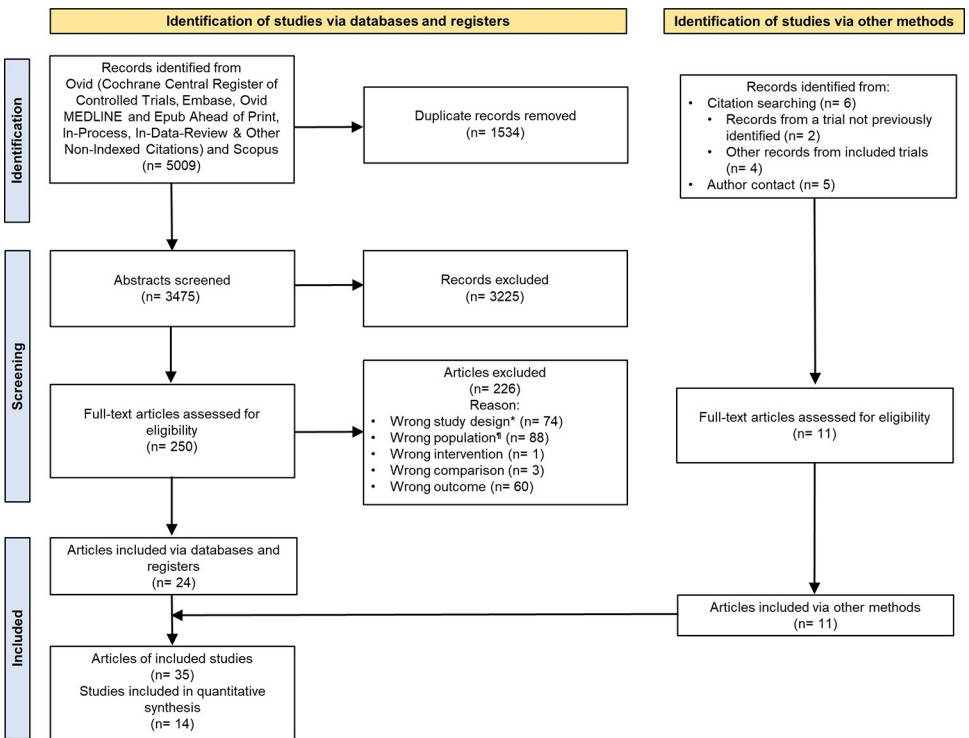

**Fig 1. PRISMA Flow chart.** *Wrong study design includes articles of rationale and protocols; ¶Wrong population includes articles of studies that analyzed subgroups of the trial sample that were not of interest for our review.

## Risk of bias

The overall risk of bias was judged as 'low' in eight trials, 'some concerns' in six trials, and no trials were judged as 'high' risk of bias. Of the trials with 'some concerns', five of them had unclear information in the selection of the reported result and one had deviations from intended intervention. Additionally, the CANVAS Program did not make an adequate statement about missing outcome data neither in any of publications nor the protocol (S1 Fig).

## Quality of the evidence

The 'Summary of the evidence' table showed moderate certainty of evidence for every outcome. Indirectness was deemed as 'serious' for every outcome, as most of the studies were designed to evaluate diseases other than CKD such as T2DM and HF (S4 Table).

## Main analysis

**Primary outcomes.** SGLT-2 inhibitors decreased the hazard for the primary CV outcome in patients with CKD regardless of the presence of T2DM (HR 0.76; [95% CI 0.72–0.79]; $\tau^2 = 0$; $I^2 = 0\%$). Likewise, SGLT-2 inhibitors significantly reduced the occurrence of the primary renal outcome (HR 0.69; [95% CI 0.61–0.79]; $\tau^2 = 0.0093$; $I^2 = 23\%$) (Fig 2).

**Secondary outcomes.** In patients with CKD with or without T2DM, SGLT-2 inhibitors decreased the risk for all-cause mortality (HR 0.87; [95% CI 0.79–0.94]; $\tau^2 = 0$; $I^2 = 0\%$), CV death (HR 0.86; [95% CI 0.81–0.91]; $\tau^2 = 0$; $I^2 = 0\%$), and hospitalization for HF (HR 0.67; [95% CI 0.62–0.73]; $\tau^2 = 0$; $I^2 = 0\%$) (Fig 3). However, SGLT-2 inhibitors did not reduce the risk for the MACE outcome (HR 0.86; [95% CI 0.74–1.01]; $\tau^2 = 0.0178$; $I^2 = 41\%$).

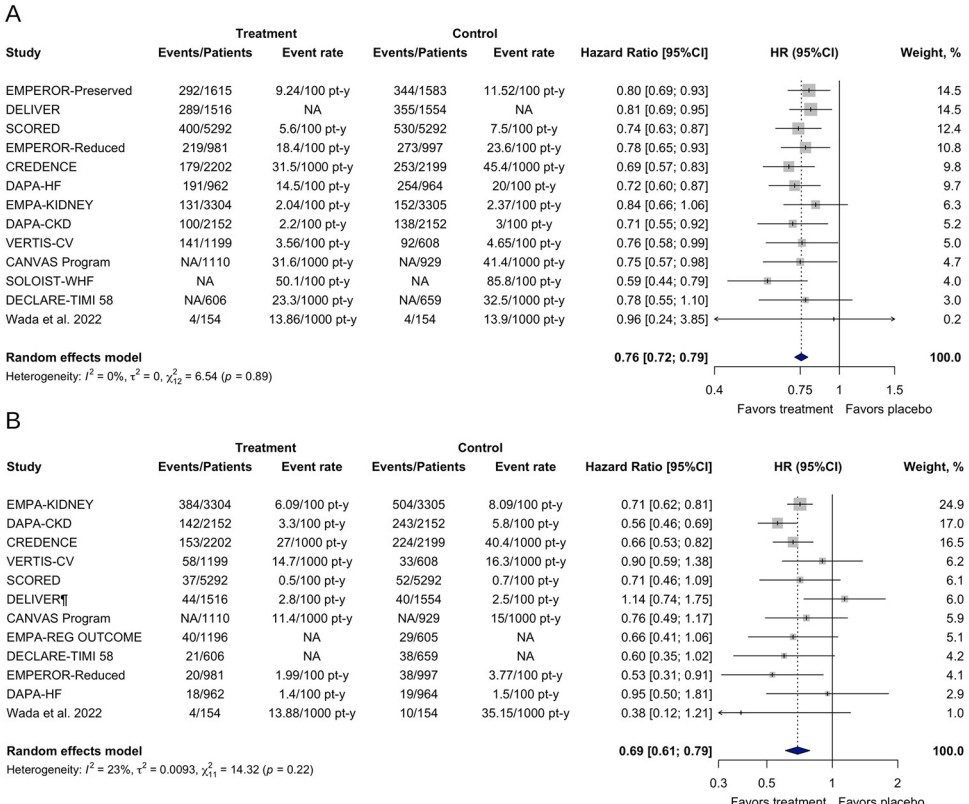

**Fig 2. Effects of SGLT-2 inhibitors on the primary outcomes in patients with CKD with or without T2DM.** (A) Primary cardiovascular outcome. (B) Primary renal outcome. ¶Effect estimate calculated with a fixed effects model meta-analysis. CI, confidence interval; HR, hazard ratio; NA, not available.

## Subgroup analysis

The subgroup analyses according to the presence of T2DM are shown in Figs 4 and 5. We did not find significant differences between diabetic and nondiabetic patients in any outcome. We found a significant reduction of the hazard for the primary renal outcome in the subgroup of patients with T2DM (HR 0.65; [95% CI 0.59–0.72]; $\tau^2 = 0$; $I^2 = 0\%$). Although this protective effect was also found in the subgroup of patients without T2DM, this association was not significant, and pooled studies were heterogenous (HR 0.66; [95% CI 0.04–11.16]; $\tau^2 = 0.0764$; $I^2 = 75\%$).

We performed a subgroup analysis according to eGFR levels (S2–S4 Figs), which showed that SGLT-2 inhibitors had a protective effect for the primary and secondary outcomes regardless of baseline eGFR, as the test for subgroup differences was not significant. Likewise, we did not find differences in our subgroup analysis by the type of SGLT-2 inhibitor in any outcome (S5–S7 Figs).

## Sensitivity analysis

We decided to perform the subgroup analysis for the primary renal outcome without the Hartung-Knapp adjustment, as it seemed to artificially increase the confidence interval. We found a narrower but still not statistically significant confidence interval for this subgroup (HR 0.66; [95% CI 0.43–1.02]; $\tau^2 = 0.0764$; $I^2 = 75\%$), and the test for subgroup differences remained not significant ($p = 0.93$).

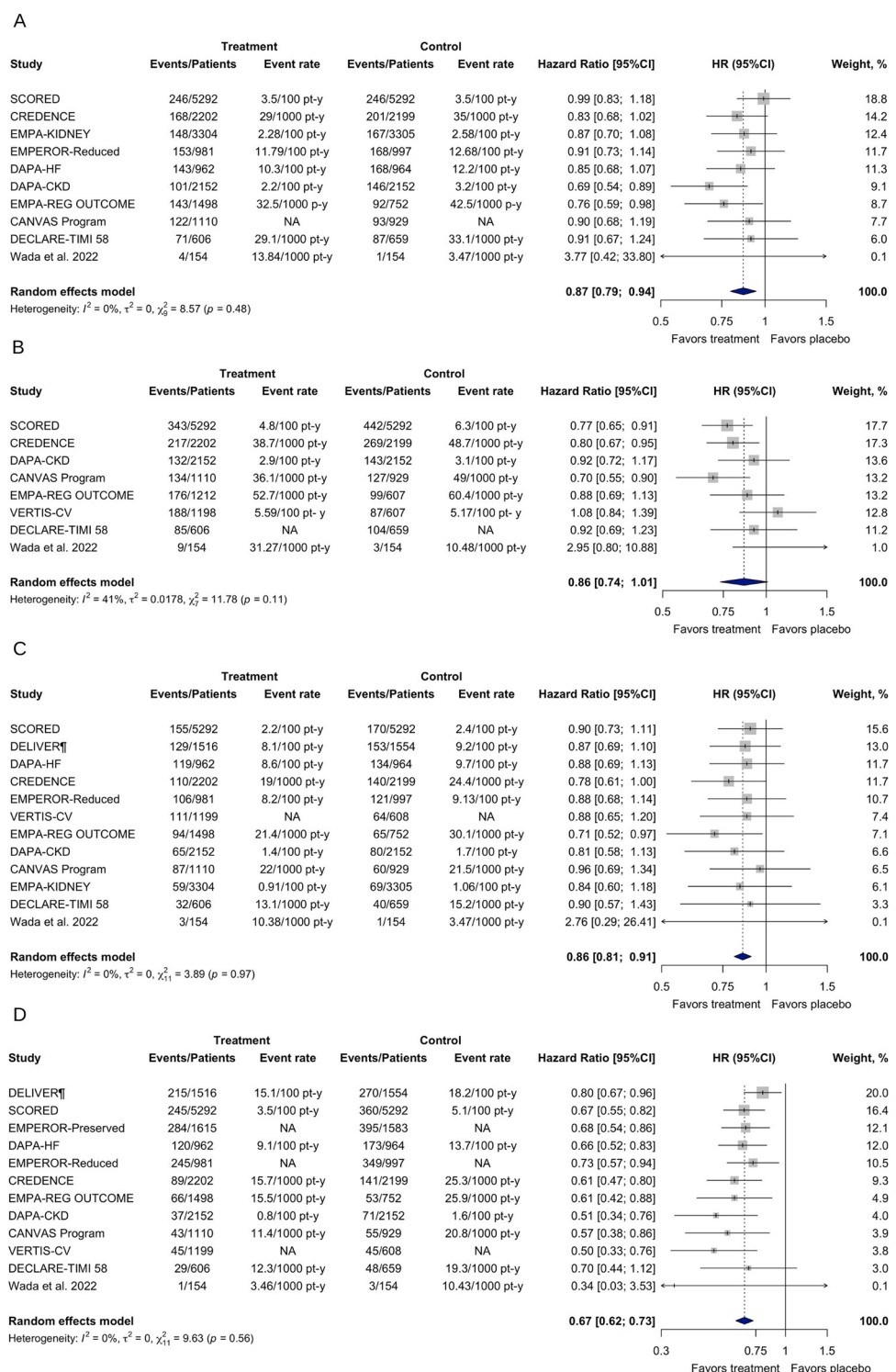

**Fig 3. Effects of SGLT-2 inhibitors on the secondary outcomes in patients with CKD with or without T2DM.** (A) All-cause mortality. (B) MACE. (C) CV death. (D) Hospitalization for HF. ¶Synthetic estimate created with a fixed effects model meta-analysis. CI, confidence interval; HR, hazard ratio; NA, not available.

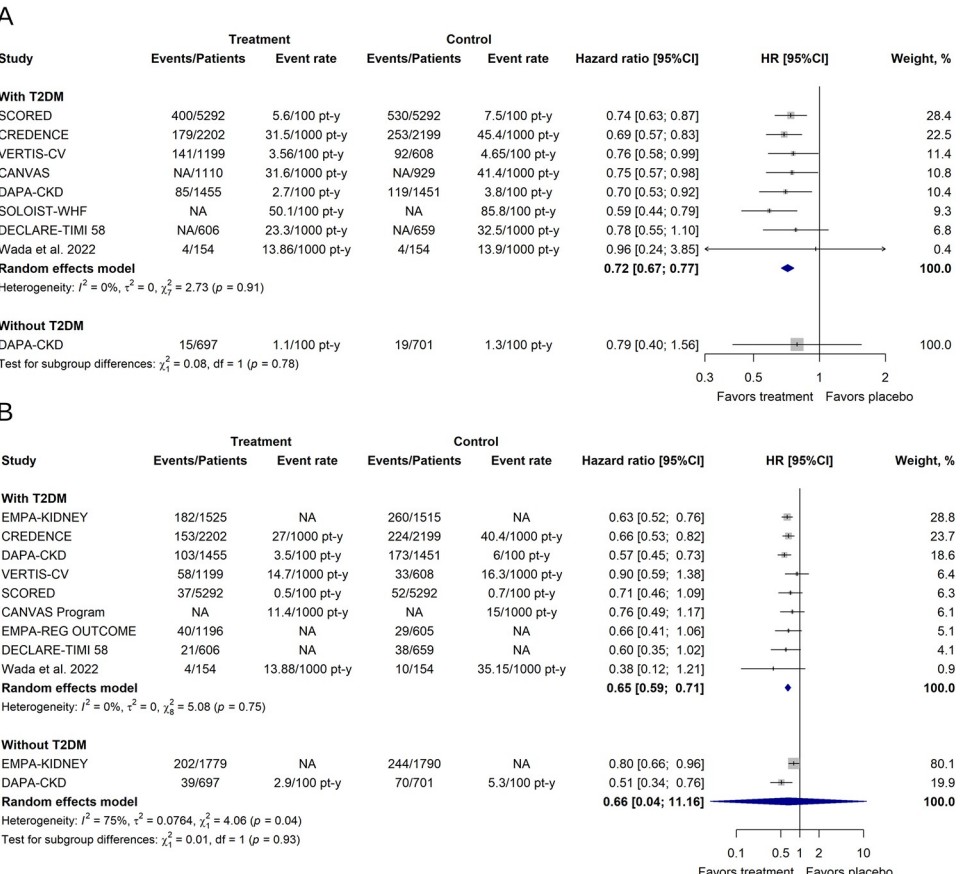

**Fig 4. Effects of SGLT-2 inhibitors on the primary outcomes in patients with CKD according to T2DM status.** (A) Primary cardiovascular outcome. (B) Primary renal outcome. CI, confidence interval; HR, hazard ratio; NA, not available; T2DM, type 2 diabetes mellitus.

The effect of SGLT-2 inhibitors when only considering patients with an eGFR < 60 ml/min/1.73 m$^2$ were consistent with the main analyses for the primary and secondary outcomes, except for the MACE outcome. The results of this sensitivity analysis are presented in the Supplement (S8 and S9 Figs).

## Publication bias assessment

We only found a significant Egger's test in the MACE and hospitalization for HF outcomes ($p < 0.05$) (S10 Fig). With regard to the availability of data, we were only able to extract data of patients with CKD without T2DM from 2 studies [10,11]. The other studies that included patients with CKD without T2DM did not report outcome data for this specific subgroup.

## Discussion

In this systematic review and meta-analysis, moderate quality of evidence shows that SGLT-2 inhibitors administered to patients with CKD with or without T2DM reduce the risk of the primary cardiovascular and renal outcomes, as well as the hazard for all-cause mortality, CV death and hospitalization for HF outcomes.

Our results are consistent with systematic reviews that focused on patients with T2DM [12–15] and patients without T2DM [63]. Other reviews included both patient groups

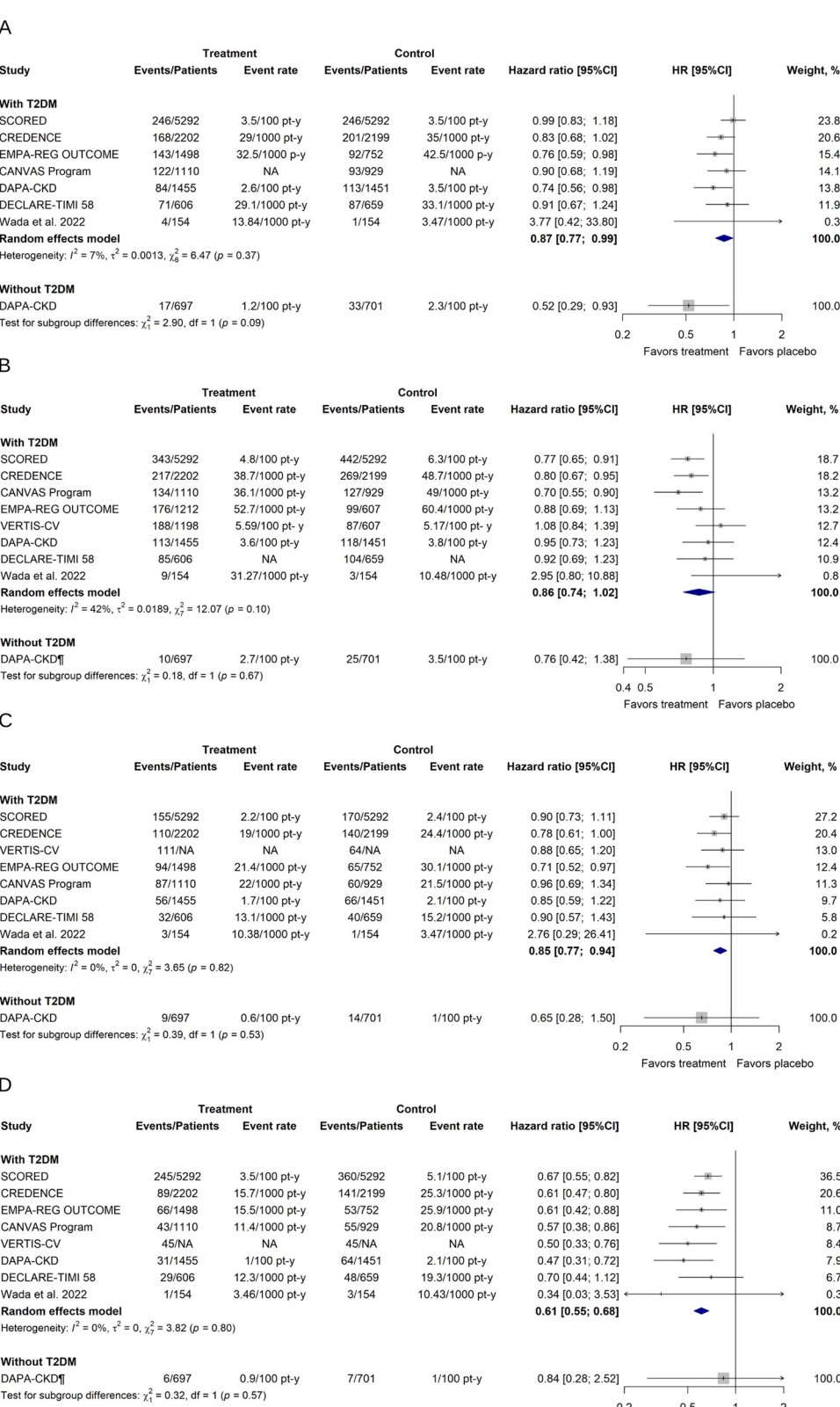

**Fig 5. Effects of SGLT-2 inhibitors on the secondary outcomes in patients with CKD according to T2DM status.**
(A) All-cause mortality. (B) MACE outcome. (C) CV death. (D) Hospitalization for HF. ¶Synthetic estimate created with a fixed effects model meta-analysis. CI, confidence interval; HR, hazard ratio; NA, not available; T2DM, type 2 diabetes mellitus.

[16,60,64,65]; however, they had limitations, such as not including patients without T2DM in the subgroup analysis of CKD patients [64], not performing a statistical test for subgroup differences [60], focusing solely on a single clinically relevant outcome [65], and not including CKD patients from trials in which they were a subgroup [16]. In contrast, our approach included patients with CKD from all trials, even those trials in which CKD patients were a subgroup of the study.

By adopting this methodological approach, we increased the number of participants with CKD from 25 898 to 44 593, when compared to the study by Baigent et al. [16]. Additionally, we noted that in some subgroup analyses of their study, the "No diabetes" subgroup was composed predominantly of non-diabetic patients with HF rather than with CKD. Consequently, the statistical test for heterogeneity by diabetes status was more reliant on data from HF patients without T2DM than CKD patients without T2DM. In our subgroup analyses, we exclusively used data from CKD patients, allowing us to assess the heterogeneity in the effect of SGLT-2 inhibitors in non-diabetic CKD patients more directly.

The subgroup analysis that assesses the differences between patients with and without T2DM did not reveal any significant subgroup effect for any outcome. This suggests that the presence of T2DM does not modify the therapeutic effect of SGLT-2 inhibitors in patients with CKD. However, there were insufficient studies reporting information for patients without T2DM. Among the six trials that included patients with CKD and with or without T2DM [10,11,32,38,56,58], only the DAPA-CKD trial provided data for every outcome in CKD patients without T2DM [10,45,54], while the EMPA-KIDNEY trial only reported such data for our primary renal outcome. Furthermore, we found high heterogeneity between studies in the subgroup of patients without T2DM for the primary renal outcome. This may be explained by multiple factors, such as differences in inclusion criteria, sample size and outcome definition; the use of different SGLT-2 inhibitors (dapagliflozin and empagliflozin); varying eGFR levels; and baseline medication. Nevertheless, our subgroup analyses by baseline eGFR values and by the type of SGLT-2 inhibitor revealed no significant differences between the subgroups. This suggests that the heterogeneity introduced by these factors does not substantially impact the overall positive effect of SGLT-2 inhibitors in CKD patients.

Our main results on the primary CV outcome were similar to those reported in previous systematic reviews [14–16,60,63]. However, our results exhibit a narrower confidence interval than those previously described. Presumably, this difference is attributable to the inclusion of new SGLT-2 inhibitors trials in patients with CKD [10,11,33,40,52] that have been published since the publication of the other systematic reviews. Thus, we analyzed a greater number of patients with CKD, and this was reflected by a higher precision in our estimates.

We found a significant reduction of the hazard for the primary renal outcome in patients with CKD and the diabetic subgroup. Our result was expected as SGLT-2 inhibitors help control T2DM, which is a major risk for the development and progression of CKD [66]. This finding is also consistent with previous systematic reviews in the diabetic population [13,16,41,64].

On the other hand, we did not find a significant hazard reduction for the renal outcome in the subgroup of patients without T2DM. This can be attributed to the small number of studies (n = 2) included in this subgroup and the substantial heterogeneity between these two studies [10,11], which typically results in a low precision of the estimated between-study variance when using the random-effects model [67]. In contrast, another systematic review [16] analyzed the data from patients with CKD without T2DM from the same clinical trials [10,11] and found a significant protective effect. We attribute this difference to two key factors. First, they redefined the renal composite outcome to include a sustained ≥50% decline in eGFR, which decreased the total number of events; and second, they used different statistical methods of analysis. Ultimately, the effect does not appear to be robust, as its statistical significance seems dependent on the method of analysis used.

The favorable effect of SGLT-2 inhibitors extends to certain secondary outcomes. However, our analysis revealed varied effects compared to existing literature. For the all-cause mortality outcome, we found a significant protective effect in CKD patients, similarly to Arnott et al. [15] but in contrast to other reviews [13,60]. Our results align with a review [14] that did not find a protective effect for the MACE outcome in CKD patients with T2DM, while other reviews [12,13,15] have shown otherwise. Regarding the cardiovascular death outcome, we found a significant protective effect in CKD patients with or without T2DM, and in those with CKD and T2DM. Two systematic reviews [15,60] support this result, but others do not [13,14,64]. Additionally, we found that SGLT-2 inhibitors have a strong protective effect for the hospitalization for HF outcome, which is consistent with prior systematic reviews [13,15,60,64]. This variability in results across systematic reviews is mostly explained by the gradual increase of published clinical trials in patients with CKD over time.

Notably, the focus of clinical trials of SGLT-2 inhibitors efficacy has shifted from patients with T2DM and CV disease to patients with HF and CKD regardless of the presence of T2DM. This can be attributed to the findings of preclinical studies that revealed that SGLT-2 inhibitors have other cardiorenal protective mechanisms independent of their glucose lowering effect, such as the reduction of glomerular hyperfiltration [6], tubular cell fibrosis and inflammation [7,8], and albuminuria [9]. The shift to study the clinical relevance of these mechanisms has had an impact on the absolute incidence of time-to-event outcomes in each clinical trial. For instance, a higher number of patients developed the primary CV outcome in the HF trials, which accounts for the higher weight these trials received in our meta-analyses. This may be explained by the fact that all the patients we selected from those trials had both HF and CKD, while only 10.9% of patients in the DAPA-CKD trial had HF and only 26.7% of patients in the EMPA-KIDNEY trial had established cardiovascular disease. Thus, CKD patients from HF trials were more susceptible to clinically unfavorable outcomes than patients from CKD trials.

This study represents a significant contribution to the field, as to the best our knowledge, this is the first systematic review that evaluated patients with CKD according to the definition of each trial, which often considered the presence of albuminuria. This approach allowed us to include patients with CKD with a preserved eGFR but a high albuminuria. Previous systematic reviews [12–15,60,64] established an eGFR $< 60$ ml/min/1.73 m$^2$ as the sole criterion for the diagnosis of CKD, potentially excluding data from patients with CKD that do not meet this threshold. A key strength of this systematic review is its comprehensive inclusion of CKD patients from every trial in the literature, even those from trials focused on other diseases such as T2DM or HF. Our approach greatly augmented our sample size and may increase the external validity of our estimates. To assess the robustness of our results, we conducted a sensitivity analysis only including patients with an eGFR $< 60$ ml/min/1.73 m$^2$. Additionally, we performed subgroup analyses according to baseline eGFR values and the type of SGLT-2 inhibitor. Finally, while there was some variability in the individual components of the composite kidney outcome in each trial, they mostly comprised clinically relevant markers such as doubling of serum creatinine, strong decline of eGFR, end-stage kidney disease, or renal death.

We recognize several limitations in our study. First, most of the clinical trials in our analysis considered the patients with CKD as a subgroup rather than as their main study population. Thus, these studies were not originally designed to detect the treatment effect in these patients. Second, some baseline and outcome data from the trials were missing. Consequently, we could not incorporate these elements into our results. For instance, we were unable to assess the effect of baseline hypoglycemic drugs in a meta-analysis, as comparisons according to this variable were scarcely reported in the records of the included trials. Third, we could only extract outcome data of patients with CKD without T2DM from two trials, which severely limited the external validity of our subgroup analyses. Finally, we did not evaluate the effect of SGLT-2

inhibitors according to specific causes of non-diabetic CKD, as we considered that the number of trials that reported this information was very low (n = 2).

In conclusion, our analysis demonstrates that SGLT-2 inhibitors, in addition to standard therapy, provide significant protection against cardiovascular and renal outcomes in CKD patients, regardless of their T2DM status. This finding aligns with the well-established roles of empagliflozin and dapagliflozin in patients with HF without diabetes [68], the indication of dapagliflozin for CKD patients without T2DM [69], and the recent findings from the EMPA-KIDNEY trial that show significant benefits of empagliflozin in CKD patients without T2DM [11]. The potential benefits of other SGLT-2 inhibitors in this context should be individually assessed in future clinical trials. Additionally, we suggest that future trials use a standardized renal composite outcome to facilitate comparisons between studies.

## Supporting information

**S1 Checklist. PRISMA 2020 checklist.**
(DOCX)

**S1 Fig. Risk of bias assessment.** (A) Traffic light plot. (B) Bar plot.
(TIF)

**S2 Fig. Effects of SGLT-2 inhibitors on the primary outcomes in patients with CKD according to eGFR.** (A) Primary cardiovascular outcome. (B) Primary renal outcome. *Included patients with an eGFR of 25 to < 45 ml/min/1.73 m$^2$. ¶Synthetic estimate created with a fixed effects model meta-analysis. CI, confidence interval; eGFR, estimated glomerular filtration rate; HR, hazard ratio; NA, not available.
(TIF)

**S3 Fig. Effects of SGLT-2 inhibitors on two secondary outcomes in patients with CKD according to eGFR.** (A) All-cause mortality. (B) MACE outcome. ¶Synthetic estimate created with a fixed effects model meta-analysis. CI, confidence interval; eGFR, estimated glomerular filtration rate; HR, hazard ratio; NA, not available.
(TIF)

**S4 Fig. Effects of SGLT-2 inhibitors on two secondary outcomes in patients with CKD according to eGFR.** (A) CV death. (B) Hospitalization for HF. *Included patients with an eGFR of 25 to < 45 ml/min/1.73 m$^2$. ¶Synthetic estimate created with a fixed effects model meta-analysis. CI, confidence interval; eGFR, estimated glomerular filtration rate; HR, hazard ratio; NA, not available.
(TIF)

**S5 Fig. Effects of SGLT-2 inhibitors on the primary outcomes in patients with CKD according to the type of SGLT-2 inhibitor.** (A) Primary cardiovascular outcome. (B) Primary renal outcome. ¶Synthetic estimate created with a fixed effects model meta-analysis. CI, confidence interval; HR, hazard ratio; NA, not available.
(TIF)

**S6 Fig. Effects of SGLT-2 inhibitors on two secondary outcomes in patients with CKD according to the type of SGLT-2 inhibitor.** (A) All-cause mortality. (B) MACE outcome. CI, confidence interval; HR, hazard ratio; NA, not available.
(TIF)

**S7 Fig. Effects of SGLT-2 inhibitors on two secondary outcomes in patients with CKD according to the type of SGLT-2 inhibitor.** (A) CV death. (B) Hospitalization for HF.

¶Synthetic estimate created with a fixed effects model meta-analysis. CI, confidence interval; HR, hazard ratio; NA, not available.
(TIF)

**S8 Fig. Sensitivity analysis: Effects of SGLT-2 inhibitors on the primary outcomes in patients with CKD and an eGFR < 60 ml/min/1.73 m$^2$.** (A) Primary cardiovascular outcome. (B) Primary renal outcome. ¶Synthetic estimate created with a fixed effects model meta-analysis. CI, confidence interval; CKD, chronic kidney disease; HR, hazard ratio; NA, not available.
(TIF)

**S9 Fig. Sensitivity analysis: Effects of SGLT-2 inhibitors on the secondary outcomes in patients with CKD and an eGFR < 60 ml/min/1.73 m$^2$.** (A) All-cause mortality. (B) MACE outcome. (C) CV death. (D) Hospitalization for HF. ¶Synthetic estimate created with a fixed effects model meta-analysis. CI, confidence interval; CKD, chronic kidney disease; HR, hazard ratio; NA, not available.
(TIF)

**S10 Fig. Reporting bias assessment: Funnel plots and Egger's tests.** (A) Primary cardiovascular outcome. (B) Primary renal outcome. (C) All-cause mortality. (D) MACE outcome. (E) CV death. (F) Hospitalization for HF.
(TIF)

**S1 Table. Definition of chronic kidney disease and the renal and cardiovascular composite outcomes in each study.** *We used an eGFR < 60 ml/min/1.73 m$^2$ as the definition of chronic kidney disease. CKD, chronic kidney disease; CV, cardiovascular; eGFR, estimated glomerular filtration rate; HF, heart failure; NA, not available; UACR, urine albumin-to-creatinine ratio.
(DOCX)

**S2 Table. Search Strategy.**
(DOCX)

**S3 Table. Clinical trial baseline characteristics ordered according to publication date.** Data calculated from other publications of the included trial is shaded blue, since it was not directly available in the published articles. ACEI, angiotensin-converting enzyme inhibitors; ARB, angiotensin receptor blockers; CKD, chronic kidney disease; CV, cardiovascular; DPP-4, dipeptidyl peptidase-4; eGFR, estimated glomerular filtration rate; GLP-1, glucose-like peptide-1; HF, Heart failure; IQR, interquartile range; MRA, mineralocorticoid receptor antagonist; n, absolute frequency; NA, not available; SD, standard deviation; SGLT-2, sodium-glucose cotransporter 2; T2DM, type 2 diabetes mellitus; UACR, urine albumin-to-creatinine ratio.
(DOCX)

**S4 Table. Summary of the evidence.**
(DOCX)

**S1 Database.**
(XLSX)

## Acknowledgments

We thank Oscar J Ponce M.D. (O.J.P.—Knowledge and Evaluation Research Unit, Mayo Clinic, USA) for helping in the development of the research protocol, as well as in the study

selection process and data analysis. We thank Paolo Wong M.D. (Universidad de Piura, Peru) for comments on the manuscript.

## Author Contributions

**Conceptualization:** Carlos Ignacio Reyes-Farias, Marcelo Reategui-Diaz, Franco Romani-Romani.

**Data curation:** Carlos Ignacio Reyes-Farias, Marcelo Reategui-Diaz, Larry Prokop.

**Formal analysis:** Carlos Ignacio Reyes-Farias, Marcelo Reategui-Diaz.

**Investigation:** Carlos Ignacio Reyes-Farias, Marcelo Reategui-Diaz.

**Methodology:** Carlos Ignacio Reyes-Farias, Marcelo Reategui-Diaz, Franco Romani-Romani, Larry Prokop.

**Project administration:** Carlos Ignacio Reyes-Farias, Marcelo Reategui-Diaz.

**Resources:** Carlos Ignacio Reyes-Farias, Marcelo Reategui-Diaz.

**Software:** Carlos Ignacio Reyes-Farias, Marcelo Reategui-Diaz, Larry Prokop.

**Supervision:** Franco Romani-Romani.

**Validation:** Carlos Ignacio Reyes-Farias, Marcelo Reategui-Diaz.

**Visualization:** Carlos Ignacio Reyes-Farias, Marcelo Reategui-Diaz.

**Writing – original draft:** Carlos Ignacio Reyes-Farias, Marcelo Reategui-Diaz, Franco Romani-Romani.

**Writing – review & editing:** Carlos Ignacio Reyes-Farias, Marcelo Reategui-Diaz, Franco Romani-Romani, Larry Prokop.

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
