## [Decision Letter · Decision Letter 0]

7 Aug 2023

PONE-D-23-13221The effect of sodium-glucose cotransporter 2 inhibitors in patients with chronic kidney disease with or without type 2 diabetes mellitus on cardiovascular and renal outcomes: a systematic review and meta-analysisPLOS ONE

Dear Dr. Reategui-Diaz,

Thank you for submitting your manuscript to PLOS ONE. After careful consideration, we feel that it has merit but does not fully meet PLOS ONE’s publication criteria as it currently stands. Therefore, we invite you to submit a revised version of the manuscript that addresses the points raised during the review process.

In particular, given what is known from the Lancet publication (2022), which is a meta-analysis that only analysed data from trials that included CKD patients and did not consider data from trials where CKD patients were a subgroup, it is unclear if conducting an additional meta-analysis is necessary. Please justify whether the current meta-analysis adds any significant additional knowledge or clinical relevance. 

We look forward to receiving your revised manuscript.

Kind regards,

Edward Zimbudzi

Academic Editor

PLOS ONE

Journal Requirements:

Reviewers' comments:

Reviewer's Responses to Questions

**Comments to the Author**

1. Is the manuscript technically sound, and do the data support the conclusions?

Reviewer #1: Yes

Reviewer #2: Yes

2. Has the statistical analysis been performed appropriately and rigorously? 

Reviewer #1: Yes

Reviewer #2: Yes

3. Have the authors made all data underlying the findings in their manuscript fully available?

Reviewer #1: Yes

Reviewer #2: Yes

4. Is the manuscript presented in an intelligible fashion and written in standard English?

Reviewer #1: Yes

Reviewer #2: Yes

5. Review Comments to the Author

Reviewer #1: I appreciate the opportunity to review the manuscript titled "The effect of sodium-glucose cotransporter 2 inhibitors in patients with chronic kidney disease with or without type 2 diabetes mellitus on cardiovascular and renal outcomes: a systematic review and meta-analysis." The authors conducted a systematic review and meta-analysis to investigate the cardiovascular and renal effects of SGLT-2 inhibitors in patients with chronic kidney disease (CKD) with and without type 2 diabetes mellitus (T2DM). The authors included not only clinical trials specifically focused on patients with CKD, but also trials dedicated to heart failure (HF) or T2DM that included CKD patients as a subgroup. I acknowledge the authors' time and effort in conducting this meta-analysis and believe that the manuscript has been improved through previous revision rounds. Below are my comments for the authors.

[1] In the Introduction section, the authors mention a recent meta-analysis study published in Lancet (Ref #16: Baigent C, Lancet 2022; 400: 1788–801). The authors state that this meta-analysis only analyzed data from trials that included CKD patients and did not consider data from trials where CKD patients were a subgroup. Therefore, the efficacy of SGLT-2 inhibitors in CKD subgroups with or without T2DM was not evaluated across all trials. It is true that the Lancet meta-analysis focused on combined outcomes from CKD trials (such as CREDENCE, SCORED, DAPA-CKD, and EMPA-KIDNEY trials), it did not include outcomes of CKD patients from trials dedicated to T2DM or HF (such as DECLARE-TIMI 58, CANVAS Program, VERTIS CV, EMPA-REG OUTCOME, DAPA-HF, EMPEROR-REDUCED, EMPEROR-PRESERVED, DELIVER, and SOLOIST-WHF). However, although the Lancet meta-analysis did not provide patient-level analysis, it is unclear if conducting an additional meta-analysis is necessary. Most trials dedicated to T2DM or HF have already shown consistent benefits of SGLT-2 inhibitors across various glomerular filtration rate (GFR) subgroups, and trials dedicated to CKD (DAPA-CKD and EMPA-KIDNEY) have demonstrated the benefits of SGLT-2 inhibitors regardless of the presence of T2DM. Therefore, it can be expected that including the CKD subgroup from trials dedicated to T2DM or HF would not significantly affect the outcomes. I recommend that the authors provide more reasonable background information in the Introduction section, focusing on why the current meta-analysis is required and clinically relevant. In my opinion, considering the consistent benefits shown in the aforementioned trials and the recently published meta-analysis in Lancet, it is uncertain whether the current meta-analysis adds significant additional knowledge or clinical relevance.

[2] In the Results section (Subgroup analysis), the authors state, "Although this protective effect was also found in the subgroup of patients without T2DM, this association was not significant, and the pooled studies were heterogeneous (HR 0.66; [95% CI 0.04-11.16]; τ2 = 0.0764; I2 = 75%)." This lack of significance and heterogeneity may be primarily attributed to the limited number of trials included. As explained by the authors in the Results section (Baseline characteristics), the DAPA-CKD trial and the EMPA-KIDNEY trial were the only sources of outcome data for the subgroup of patients with CKD but without T2DM. Moreover, there was significant heterogeneity between these two trials (DAPA-CKD vs. EMPA-KIDNEY trial). Unfortunately, this limitation is significant because the results from patients with CKD but without T2DM would be a crucial finding in terms of the novelty of the current meta-analysis.

[3] The authors rated the certainty of evidence for MACE and HF hospitalization outcomes as "low-quality evidence" and mentioned concerns of indirectness and publication bias. The suggestion of possible publication bias was based on the Egger's test, as shown in Supplementary Figure S4. However, it is important to note that funnel plot asymmetry can have causes other than publication bias. Additionally, the varying number of patients from each included trial should be considered, as certain trials were dedicated to CKD patients, resulting in larger numbers of patients with CKD compared to trials dedicated to DM or HF. I recommend that the authors provide additional reasons for rating the evidence as "low quality" for the MACE and HF hospitalization outcomes.

[4] Although the authors mention in the Limitation section, it would be interesting to explore and compare the outcomes between individual SGLT2 inhibitors. This could be considered as an additional analysis within the current study or as a separate study.

[5] Was there any difference in terms of GFR values? Since many trials conducted subgroup analyses based on GFR values (e.g., GFR 45-60 vs. GFR <45), it would enhance the novelty and relevance of the current study if the authors could provide subgroup analyses based on GFR values.

Reviewer #2: 1. “Finally, only two studies compared patients with CKD and T2DM with patients with CKD without T2DM.” The above sentence is misexpressed and prone to ambiguity.

2. When performing subgroup analysis, why chronic kidney disease was not grouped according to grade of kidney function, so as to determine the different levels of benefit of SGLT-2 inhibitors in chronic kidney disease.

3. The concept of chronic kidney disease may also include diabetic nephropathy, which in turn includes type 1 diabetic nephropathy and type 2 diabetic nephropathy. Are these concepts consistent with the concept of chronic kidney disease combined with diabetes in this study?

4. As the first systematic review that to evaluat patients with CKD according to the definition of each trial, which often considered the presence of albuminuria. It is indeed suggested that the clinical rational selection of hypoglycemic drugs, especially the hypoglycemic methods of patients with kidney disease are more optimized.

5. The inconsistence in co-use of hypoglycemic agents in patients enrolled in each RCT clinical study, and the uncertainty about whether patients were treated with nephrotic agents, should explain the bias caused by this factor.

6. PLOS authors have the option to publish the peer review history of their article (what does this mean?). If published, this will include your full peer review and any attached files.

Reviewer #1: No

Reviewer #2: None

---

## [Author Response · Author response to Decision Letter 0]

16 Sep 2023

Reviewer #1: 

[1] In the Introduction section, the authors mention a recent meta-analysis study published in Lancet (Ref #16: Baigent C, Lancet 2022; 400: 1788–801). The authors state that this meta-analysis only analyzed data from trials that included CKD patients and did not consider data from trials where CKD patients were a subgroup. Therefore, the efficacy of SGLT-2 inhibitors in CKD subgroups with or without T2DM was not evaluated across all trials. It is true that the Lancet meta-analysis focused on combined outcomes from CKD trials (such as CREDENCE, SCORED, DAPA-CKD, and EMPA-KIDNEY trials), it did not include outcomes of CKD patients from trials dedicated to T2DM or HF (such as DECLARE-TIMI 58, CANVAS Program, VERTIS CV, EMPA-REG OUTCOME, DAPA-HF, EMPEROR-REDUCED, EMPEROR-PRESERVED, DELIVER, and SOLOIST-WHF). However, although the Lancet meta-analysis did not provide patient-level analysis, it is unclear if conducting an additional meta-analysis is necessary. Most trials dedicated to T2DM or HF have already shown consistent benefits of SGLT-2 inhibitors across various glomerular filtration rate (GFR) subgroups, and trials dedicated to CKD (DAPA-CKD and EMPA-KIDNEY) have demonstrated the benefits of SGLT-2 inhibitors regardless of the presence of T2DM. Therefore, it can be expected that including the CKD subgroup from trials dedicated to T2DM or HF would not significantly affect the outcomes. I recommend that the authors provide more reasonable background information in the Introduction section, focusing on why the current meta-analysis is required and clinically relevant. In my opinion, considering the consistent benefits shown in the aforementioned trials and the recently published meta-analysis in Lancet, it is uncertain whether the current meta-analysis adds significant additional knowledge or clinical relevance. 

Response: We appreciate your comments on our manuscript. We deeply respect the thoroughness of the research done by Baigent et al. However, by focusing on patients with CKD, we believe that our study possesses methodological advantages that contribute to the strengthening of the evidence for the use of SGLT-2 inhibitors specifically in patients with CKD, regardless of the presence of T2DM: 

Our decision to include every published trial in our analysis, even those in which CKD patients were a subgroup of the study, allowed us to include a considerably larger sample of patients. We included 44 593 patients with CKD, almost double the number included by Baigent et al. (25 898 patients). Thus, the number of CKD patients who come from subgroups of T2DM and HF trials is substantial, and their clinical outcome data contributes to the meta-analysis, possibly giving it more robustness. Additionally, the inclusion of every trial allowed us to better identify and show the limitations of current literature, such as the heterogeneity in the definition of the renal composite outcome across trials or the lack of reporting of various cardiorenal outcomes in patients without T2DM from the EMPA-KIDNEY trial. 

We assessed the heterogeneity between CKD patients with and without T2DM with a statistical test for every outcome, in order to confirm the absence of statistical differences between subgroups. Baigent et al. only performed this test in the overall sample of the trials, which included patients with and without CKD. This limitation is present in their Figure 3 (https://doi.org/10.1016/S0140-6736(22)02074-8). In their analysis, the subgroup “No diabetes” is further categorized in “Stable heart failure trials” and “Chronic kidney disease trials”. 

By including “Stable heart failure trials” in the “No diabetes” subgroup, the positive effect of SGLT-2 inhibitors in patients from “Chronic kidney disease trials” may have been obscured by the already established benefits of these drugs in patients with HF but neither CKD nor T2DM. For instance, the outcome “Cardiovascular death or hospitalization for heart failure” shows a wide and non-significant confidence interval for “Chronic kidney disease trials” and gives more weight to the “Stable heart failure trials”. By doing so, the test for “Heterogeneity by diabetes status” is far more reliant on the data from patients with HF without T2DM than patients with CKD without T2DM. This limitation is present in the four outcomes shown in their Figure 3. Thus, it is uncertain whether there is heterogeneity between CKD patients with and without T2DM in these clinically relevant outcomes. In consequence, a more focused analysis was warranted. 

The only analyses that provide a pure comparison between CKD patients with and without T2DM in the review by Baigent et al. are those reported in their Figure 2 and Supplementary Webfigure 2, and these only show the outcomes “Kidney disease progression” and “Kidney failure”, respectively. Additionally, those analyses did not include a vast number of participants with CKD from subgroups of HF and T2DM trials, as we mentioned before. On the other hand, our systematic review also evaluated kidney disease progression (“Primary renal outcome”) as well as other clinically relevant outcomes with a focus on patients with CKD, such as cardiovascular death, hospitalization for heart failure, all-cause mortality, and MACE (Major Adverse Cardiovascular Events). Of note, the work of Baigent et al. did not report the effect of SGLT-2 on MACE in any analysis. 

Thanks to your and the other reviewer’s kind suggestions, we have included new subgroup analyses according to the eGFR values and individual SGLT-2 inhibitors. We believe that these additions have enriched our study with greater novelty and comprehensiveness. Further details regarding these new analyses are provided in their respective responses within this document, as well as in our manuscript. 

As recommended, we have improved the Introduction and Discussion of our Manuscript to reflect the ideas discussed here. 

[2] In the Results section (Subgroup analysis), the authors state, "Although this protective effect was also found in the subgroup of patients without T2DM, this association was not significant, and the pooled studies were heterogeneous (HR 0.66; [95% CI 0.04-11.16]; τ2 = 0.0764; I2 = 75%)." This lack of significance and heterogeneity may be primarily attributed to the limited number of trials included. As explained by the authors in the Results section (Baseline characteristics), the DAPA-CKD trial and the EMPA-KIDNEY trial were the only sources of outcome data for the subgroup of patients with CKD but without T2DM. Moreover, there was significant heterogeneity between these two trials (DAPA-CKD vs. EMPA-KIDNEY trial). Unfortunately, this limitation is significant because the results from patients with CKD but without T2DM would be a crucial finding in terms of the novelty of the current meta-analysis. 

Response: We appreciate the comment, and we agree with you. We believe that the high heterogeneity between the DAPA-CKD and the EMPA-KIDNEY trials, which leads to a wide confidence interval, is indeed a limitation. However, this is not an intrinsic limitation of our study, but of the available literature, which is still lacking in trials that assess the effect of SGLT-2 inhibitors in non-diabetic patients with CKD. This is reflected in the fact that the only outcome in which we could calculate an HR for CKD patients without T2DM was the Renal composite outcome, as only the DAPA-CKD trial reported data of this subgroup for the other clinical outcomes, unlike the EMPA-KIDNEY trial. We think our study helps by drawing attention to this limitation of the literature. 

Regarding the lack of statistical significance and high heterogeneity between trials, we have commented in our Discussion section (Lines 327-344 of the Manuscript) how the selection of analytical methods may be the reason why our estimate for the kidney composite outcome differs from the one reported by Baigent et al (who obtained a statistically significant interval). This suggests that the overall effect of SGLT-2 inhibitors in the renal outcome in patients with CKD without T2DM may not be very robust, as its statistical significance seems to depend on the method of analysis used, and that more studies are needed. 

[3] The authors rated the certainty of evidence for MACE and HF hospitalization outcomes as "low-quality evidence" and mentioned concerns of indirectness and publication bias. The suggestion of possible publication bias was based on the Egger's test, as shown in Supplementary Figure S4. However, it is important to note that funnel plot asymmetry can have causes other than publication bias. Additionally, the varying number of patients from each included trial should be considered, as certain trials were dedicated to CKD patients, resulting in larger numbers of patients with CKD compared to trials dedicated to DM or HF. I recommend that the authors provide additional reasons for rating the evidence as "low quality" for the MACE and HF hospitalization outcomes. 

Response: We agree with your comment. After reviewing the GRADE handbook for the assessment of the quality of the evidence as well as the Cochrane handbook for systematic reviews (Chapter 13), we have decided that there is no indication that the evidence for the MACE and hospitalization for HF is of low quality. Thus, we have corrected it to “moderate quality”. 

We agree with you about the fact that funnel plot asymmetry and the Egger’s test are not absolute indicators of publication bias, as stated in the Cochrane Handbook (Chapter 13): “funnel plot asymmetry should not be considered to be diagnostic for the presence of non-reporting bias.” Other possible sources of this asymmetry can be poor methodological quality in smaller studies, true heterogeneity, artefactual or chance. 

In our systematic review, we believe that the inclusion of the study by Wada et al. may be a source of funnel plot asymmetry, as it was considerably smaller than the other trials and restricted to Japanese population instead of being multinational. This conclusion relies on the following statement of the Cochrane handbook (Chapter 13): “Although funnel plot asymmetry has long been equated with non-reporting bias, the funnel plot should be seen as a generic means of displaying small-study effects: a tendency for the intervention effects estimated in smaller studies to differ from those estimated in larger studies.” Additionally, the GRADE tool handbook states that “in smaller studies, over-estimate of effect may yield an asymmetric funnel plot that could be explained by limitations other than publication bias such as a restrictive study population”, which may be the case of the trial by Wada et al. 

Most of the trials included in our study report the MACE and hospitalization for HF outcomes appropriately. Consequently, there is no suggestion that these two outcomes have been intentionally underreported in published trials. Likewise, in our exhaustive literature search we did not find evidence of unpublished or unfinished studies with unfavorable results for these two outcomes that would lead to publication bias. 

We believe this change does not affect our main results, and we have corrected the Manuscript and the S4 Table to reflect our reassessment of the quality of the evidence. 

Schünemann H H, Brożek J, Guyatt G, Oxman A. GRADE handbook for grading quality of evidence and strength of recommendations. The GRADE Working Group; 2013. Available from: guidelinedevelopment.org/handbook

Higgins JPT, Thomas J, Chandler J, Cumpston M, Li T, et al. Cochrane Handbook for Systematic Reviews of Interventions version 6.4 (updated August 2023). Cochrane, 2023. Available from www.training.cochrane.org/handbook. 

[4] Although the authors mention in the Limitation section, it would be interesting to explore and compare the outcomes between individual SGLT2 inhibitors. This could be considered as an additional analysis within the current study or as a separate study. 

Response: We agree with your suggestion, and thus we have performed a new subgroup analysis according to the individual SGLT-2 inhibitors, which showed no differences across subgroups in the effect of SGLT-2 inhibitors on any outcome. The results of this analysis are shown on the new Supplementary figures S5-S7. We have commented on this finding in the Discussion section. 

[5] Was there any difference in terms of GFR values? Since many trials conducted subgroup analyses based on GFR values (e.g., GFR 45-60 vs. GFR <45), it would enhance the novelty and relevance of the current study if the authors could provide subgroup analyses based on GFR values. 

Response: We agree with your suggestion, and thus we have performed a new subgroup analysis according to eGFR values, which showed no differences across subgroups on any outcome. The results of this analysis are shown on the new Supplementary figures S2-S4. We have commented on this finding in the Discussion section. 

Of note, for this analysis we had to retrieve an additional publication (Waijer et al., 2022) of an already included study (DAPA-CKD), as it contained the data for the outcomes stratified by eGFR. This article has been included in the Results section and Flowchart as “Other records from included trials”. 

Waijer SW, Vart P, Cherney DZI, Chertow GM, Jongs N, Langkilde AM, et al. Effect of dapagliflozin on kidney and cardiovascular outcomes by baseline KDIGO risk categories: a post hoc analysis of the DAPA-CKD trial. Diabetologia. 2022;65: 1085–1097. doi:10.1007/s00125-022-05694-6 

Reviewer #2: 

1. “Finally, only two studies compared patients with CKD and T2DM with patients with CKD without T2DM.” The above sentence is misexpressed and prone to ambiguity. 

Response: We agree with your suggestion, and thus we have enhanced the clarity of the sentence. 

New version: “Finally, only two studies made comparisons between patients with CKD and T2DM and patients with CKD without T2DM”. 

 2. When performing subgroup analysis, why chronic kidney disease was not grouped according to grade of kidney function, so as to determine the different levels of benefit of SGLT-2 inhibitors in chronic kidney disease. 

Response: We agree with your suggestion, and thus we have performed a new subgroup analysis according to eGFR values, which showed no differences across subgroups on any outcome. The results of this analysis are shown on the new Supplementary figures S2-S4. We have commented on this finding in the Discussion section. 

Of note, for this analysis we had to retrieve an additional publication (Waijer et al., 2022) of an already included study (DAPA-CKD), as it contained the data for the outcomes stratified by eGFR. This article has been included in the Results section and Flowchart as “Other records from included trials”. 

Waijer SW, Vart P, Cherney DZI, Chertow GM, Jongs N, Langkilde AM, et al. Effect of dapagliflozin on kidney and cardiovascular outcomes by baseline KDIGO risk categories: a post hoc analysis of the DAPA-CKD trial. Diabetologia. 2022;65: 1085–1097. doi:10.1007/s00125-022-05694-6 

 3. The concept of chronic kidney disease may also include diabetic nephropathy, which in turn includes type 1 diabetic nephropathy and type 2 diabetic nephropathy. Are these concepts consistent with the concept of chronic kidney disease combined with diabetes in this study? 

Response: In our study, the vast majority of patients with both CKD and diabetes had specifically Type 2 diabetes mellitus. 

While in clinical practice it is true that type 1 diabetes mellitus (T1DM) may be a cause of nephropathy, patients with T1DM were excluded from almost every trial by design. The only exceptions are the EMPEROR-Reduced and EMPEROR-Preserved trials, which do not exclude patients with T1DM. Unfortunately, these trials only report the number of participants with “Diabetes” and do not specify the type of diabetes, so that the exact number of participants with T1DM in these trials is uncertain. 

However, the EMPEROR trials explicitly excluded patients with a history of diabetic ketoacidosis. Diabetic ketoacidosis is a very frequent form of initial presentation of undiagnosed T1DM in children, reported in around 30% of cases, ranging from 12 to 80% depending on the study (Usher-Smith et al., Cherubini et al.). Moreover, the incidence of diabetic ketoacidosis in patients already diagnosed with T1DM is high, at around 5 to 7% (Kao et al.). Therefore, by excluding patients with a history of diabetic ketoacidosis, the EMPEROR trials excluded a large proportion of T1DM patients. It is safe to assume that the number of patients with T1DM and nephropathy in the EMPEROR trials was rather low, and thus the impact they could have on the overall outcome estimates in our meta-analyses was probably minimal. 

We have added a comment about this in the Results – Baseline characteristics section (Lines 197-198 of the Manuscript). 

Usher-Smith JA, Thompson M, Ercole A, Walter FM. Variation between countries in the frequency of diabetic ketoacidosis at first presentation of type 1 diabetes in children: a systematic review. Diabetologia. 2012 Nov;55(11):2878-94. doi: 10.1007/s00125-012-2690-2 

Cherubini V, Grimsmann JM, Åkesson K, Birkebæk NH, Cinek O, Dovč K, et al. Temporal trends in diabetic ketoacidosis at diagnosis of paediatric type 1 diabetes between 2006 and 2016: results from 13 countries in three continents. Diabetologia. 2020;63(8):1530-1541. doi: 10.1007/s00125-020-05152-1 

Kao KT, Islam N, Fox DA, Amed S. Incidence Trends of Diabetic Ketoacidosis in Children and Adolescents with Type 1 Diabetes in British Columbia, Canada. J Pediatr. 2020;221:165-173.e2. doi: 10.1016/j.jpeds.2020.02.069 

 4. As the first systematic review that to evaluate patients with CKD according to the definition of each trial, which often considered the presence of albuminuria. It is indeed suggested that the clinical rational selection of hypoglycemic drugs, especially the hypoglycemic methods of patients with kidney disease are more optimized. 

 5. The inconsistence in co-use of hypoglycemic agents in patients enrolled in each RCT clinical study, and the uncertainty about whether patients were treated with nephrotic agents, should explain the bias caused by this factor. 

Response: We believe that your points 4 and 5 were intended to be a single point, so we have addressed both here: 

The inclusion of albuminuria as a diagnostic criterion of CKD could indeed be a factor of bias in our meta-analysis, considering that some drugs such as ACE inhibitors can influence it. Being aware of this, we decided to perform a sensitivity analysis with CKD exclusively defined by an eGFR < 60 ml/min/1.73 m2 regardless of albuminuria. The results were largely consistent with those of our main analysis, which included albuminuria in the definition of CKD. 

We agree that the inconsistency of baseline hypoglycemic drugs could be a factor of bias as well. Unfortunately, comparisons according to baseline hypoglycemic drugs are scarcely reported in the records of the included trials, making it difficult to assess the effect of this variable in a meta-analysis. This is probably due to the design of all the included trials, which allowed most patients to continue taking their usual medication. In many cases patients used more than one drug, which made subgroup comparisons unfeasible. In our search we found a single analysis focusing on this issue from the VERTIS-CV trial by Dagogo-Jack et al., which showed no treatment effect modification of ertugliflozin on cardiorenal outcomes when stratified by baseline use of metformin, insulin, sulphonylureas and dipeptidyl peptidase‐4 inhibitors. We did not find information on the other SGLT-2 inhibitors. 

We added a comment regarding this observation in the Discussion section (lines 394-397 of the Manuscript), as a limitation. 

Dagogo‐Jack S, Cannon CP, Cherney DZI, Cosentino F, Liu J, Pong A, et al. Cardiorenal outcomes with ertugliflozin assessed according to baseline glucose‐lowering agent: An analysis from VERTIS CV. Diabetes Obes Metab. 2022;24: 1245–1254. doi:10.1111/dom.14691

---

## [Decision Letter · Decision Letter 1]

11 Oct 2023

PONE-D-23-13221R1The effect of sodium-glucose cotransporter 2 inhibitors in patients with chronic kidney disease with or without type 2 diabetes mellitus on cardiovascular and renal outcomes: a systematic review and meta-analysisPLOS ONE

Dear Dr. Reategui-Diaz,

Thank you for submitting your manuscript to PLOS ONE. After careful consideration, we feel that it has merit but does not fully meet PLOS ONE’s publication criteria as it currently stands. Therefore, we invite you to submit a revised version of the manuscript that addresses the points raised during the review process.

 In particular, the discussion section in the current version of your manuscript is quite lengthy. This could be made more concise. 

We look forward to receiving your revised manuscript.

Kind regards,

Edward Zimbudzi

Academic Editor

PLOS ONE

Journal Requirements:

Reviewers' comments:

Reviewer's Responses to Questions

**Comments to the Author**

1. If the authors have adequately addressed your comments raised in a previous round of review and you feel that this manuscript is now acceptable for publication, you may indicate that here to bypass the “Comments to the Author” section, enter your conflict of interest statement in the “Confidential to Editor” section, and submit your "Accept" recommendation.

Reviewer #1: (No Response)

Reviewer #2: All comments have been addressed

2. Is the manuscript technically sound, and do the data support the conclusions?

Reviewer #1: Yes

Reviewer #2: Yes

3. Has the statistical analysis been performed appropriately and rigorously? 

Reviewer #1: Yes

Reviewer #2: Yes

4. Have the authors made all data underlying the findings in their manuscript fully available?

Reviewer #1: Yes

Reviewer #2: Yes

5. Is the manuscript presented in an intelligible fashion and written in standard English?

Reviewer #1: Yes

Reviewer #2: Yes

6. Review Comments to the Author

Reviewer #1: I would like to commend the authors for their diligent efforts in addressing the previous review comments, which have led to a significant improvement in the manuscript. Additionally, I have some further comments to offer.

[1] The Discussion section appears quite lengthy. I recommend that the authors work on making it more concise.

[2] When explaining the subgroup analysis based on the type of SGLT-2 inhibitors, please consistently use the term "the type of SGLT-2 inhibitors" instead of simply referring to "the SGLT-2 inhibitor" throughout the manuscript.

[3] In order to emphasize the significance and relevance of the current study, please indicate that this study aimed to include all the available patients with CKD from the current literature, not only those from clinical trials dedicated to CKD, but also those where CKD patients were included as a subgroup.

[4] In the third paragraph of the Introduction section (lines 57 – 61), I would like to recommend the authors to rephrase the sentences. Because the aim of the present study appears in the next paragraph, the sentence above should highlight the needs for an additional meta-analysis, while avoiding the term “our results”. For example, “However, to date, there has been no comprehensive systematic review assessing the impact of SGLT-2 inhibitors on the entire CKD population, as provided in the available literature. Incorporating data from all CKD patients within clinical trials, not solely those designed exclusively for CKD patients but also those where CKD patients are included as a subgroup, could enhance the accuracy of effect estimates and enhance the applicability of the findings.” Please note that this is just a suggestion, and I hope the authors can refine the sentences further.

[5] Likewise, in the last paragraph of the Introduction section (lines 62 – 64), consider rephrasing it as follows: "This systematic review aims to assess the cardiovascular and renal effects of SGLT-2 inhibitors in all available CKD patients in the current literature and to evaluate whether the presence of T2DM modifies their overall impact." Please note that this is just a suggestion.

[6] In the Discussion section (line 299), please specify in the sentence that the total number of participants with CKD was compared to the study by Baigent et al.

Reviewer #2: The authors responded to all of the review questions one on one and included detailed evidence. In particular, authors assessed heterogeneity between CKD patients with and without T2DM and performed statistical tests for each outcome to confirm that there were no statistical differences between subgroups.

7. PLOS authors have the option to publish the peer review history of their article (what does this mean?). If published, this will include your full peer review and any attached files.

Reviewer #1: No

Reviewer #2: **Yes: **Hengfen Dai

---

## [Author Response · Author response to Decision Letter 1]

23 Oct 2023

Dear reviewers,

Thank you for your further suggestions about our manuscript. We have implemented the changes and improved our manuscript accordingly. 

Sincerely,

Marcelo Reategui-Diaz

Reviewer #1:

[1] The Discussion section appears quite lengthy. I recommend that the authors work on making it more concise.

Response: We agree with your suggestion, and thus we have reduced the number of words from most of the Discussion section paragraphs. With these changes, the word count was reduced from 2236 to 1623 words.

[2] When explaining the subgroup analysis based on the type of SGLT-2 inhibitors, please consistently use the term "the type of SGLT-2 inhibitors" instead of simply referring to "the SGLT-2 inhibitor" throughout the manuscript.

Response: We agree with your suggestion. In the new version, we have used the term "the type of SGLT-2 inhibitors" more consistently across the manuscript.

[3] In order to emphasize the significance and relevance of the current study, please indicate that this study aimed to include all the available patients with CKD from the current literature, not only those from clinical trials dedicated to CKD, but also those where CKD patients were included as a subgroup.

Response: We agree with your suggestion. We modified the Abstract, Introduction, Methods and Discussion sections to emphasize that our study aimed to assess the impact of SGLT-2 inhibitors on the entire CKD population in the current literature, including both CKD trials and trials in which CKD patients were a subgroup.

[4] In the third paragraph of the Introduction section (lines 57 – 61), I would like to recommend the authors to rephrase the sentences. Because the aim of the present study appears in the next paragraph, the sentence above should highlight the needs for an additional meta-analysis, while avoiding the term “our results”. For example, “However, to date, there has been no comprehensive systematic review assessing the impact of SGLT-2 inhibitors on the entire CKD population, as provided in the available literature. Incorporating data from all CKD patients within clinical trials, not solely those designed exclusively for CKD patients but also those where CKD patients are included as a subgroup, could enhance the accuracy of effect estimates and enhance the applicability of the findings.” Please note that this is just a suggestion, and I hope the authors can refine the sentences further.

Response: We modified the Introduction section to incorporate your suggestions in this paragraph (lines 57-62).

[5] Likewise, in the last paragraph of the Introduction section (lines 62 – 64), consider rephrasing it as follows: "This systematic review aims to assess the cardiovascular and renal effects of SGLT-2 inhibitors in all available CKD patients in the current literature and to evaluate whether the presence of T2DM modifies their overall impact." Please note that this is just a suggestion.

Response: We agree with your suggestion. We modified the Introduction section to incorporate your suggestion in this paragraph (lines 63-65).

[6] In the Discussion section (line 299), please specify in the sentence that the total number of participants with CKD was compared to the study by Baigent et al.

Response: We agree with your suggestion. We modified the Discussion section to incorporate your suggestion in this paragraph (line 286).

Reviewer #2: 

The authors responded to all of the review questions one on one and included detailed evidence. In particular, authors assessed heterogeneity between CKD patients with and without T2DM and performed statistical tests for each outcome to confirm that there were no statistical differences between subgroups

Response: Thank you for your thorough evaluation of our manuscript.

---

## [Decision Letter · Decision Letter 2]

30 Oct 2023

PONE-D-23-13221R2The effect of sodium-glucose cotransporter 2 inhibitors in patients with chronic kidney disease with or without type 2 diabetes mellitus on cardiovascular and renal outcomes: a systematic review and meta-analysisPLOS ONE

Dear Dr. Reategui-Diaz,

Thank you for submitting your manuscript to PLOS ONE. After careful consideration, we feel that it has merit but does not fully meet PLOS ONE’s publication criteria as it currently stands. Therefore, we invite you to submit a revised version of the manuscript that addresses the points raised during the review process.

We look forward to receiving your revised manuscript.

Kind regards,

Edward Zimbudzi

Academic Editor

PLOS ONE

Journal Requirements:

**Additional Editor Comments:**

Note that our reviewers suggest reorganising paragraphs in the discussion to improve the flow of this section. I recommend that you pay particular attention to the examples provided.  

Reviewers' comments:

Reviewer's Responses to Questions

**Comments to the Author**

1. If the authors have adequately addressed your comments raised in a previous round of review and you feel that this manuscript is now acceptable for publication, you may indicate that here to bypass the “Comments to the Author” section, enter your conflict of interest statement in the “Confidential to Editor” section, and submit your "Accept" recommendation.

Reviewer #1: (No Response)

2. Is the manuscript technically sound, and do the data support the conclusions?

Reviewer #1: Yes

3. Has the statistical analysis been performed appropriately and rigorously? 

Reviewer #1: Yes

4. Have the authors made all data underlying the findings in their manuscript fully available?

Reviewer #1: Yes

5. Is the manuscript presented in an intelligible fashion and written in standard English?

Reviewer #1: Yes

6. Review Comments to the Author

Reviewer #1: I want to express my gratitude for the authors' dedication and time spent on the revision. The authors have made substantial efforts in addressing the comments from the previous revision. While I acknowledge the significant improvements in the manuscript, I still believe that the Discussion section could benefit from further refinement. Some paragraphs appear to disrupt the overall flow and readability. Specifically, the paragraph beginning with "Independently of their glucose-lowering effect..." seems inconsistent with the surrounding paragraphs. Additionally, the paragraph that starts with "To date, the role of empagliflozin and dapagliflozin..." doesn't seem to flow seamlessly with the adjacent paragraphs.

Would it be possible to consider reorganizing the paragraphs (not only the above two paragraphs I have mentioned, but also other paragraphs) within the Discussion section?

7. PLOS authors have the option to publish the peer review history of their article (what does this mean?). If published, this will include your full peer review and any attached files.

Reviewer #1: No

---

## [Author Response · Author response to Decision Letter 2]

8 Nov 2023

Dear reviewer,

Thank you for your comments about the Discussion section of our manuscript. We have implemented the changes and improved our manuscript accordingly. 

We agree with your comments. In consequence, we have sorted the paragraphs to follow a more logical order. We also paid particular attention to the two paragraphs that were highlighted in your comment. We have merged those paragraphs with adjacent paragraphs seamlessly, so that the readability and connection of ideas are improved.

---

## [Decision Letter · Decision Letter 3]

15 Nov 2023

The effect of sodium-glucose cotransporter 2 inhibitors in patients with chronic kidney disease with or without type 2 diabetes mellitus on cardiovascular and renal outcomes: a systematic review and meta-analysis

PONE-D-23-13221R3

Dear Dr. Reategui-Diaz,

We’re pleased to inform you that your manuscript has been judged scientifically suitable for publication and will be formally accepted for publication once it meets all outstanding technical requirements.

Kind regards,

Edward Zimbudzi

Academic Editor

PLOS ONE

Additional Editor Comments (optional):

Reviewers' comments:

Reviewer's Responses to Questions

**Comments to the Author**

1. If the authors have adequately addressed your comments raised in a previous round of review and you feel that this manuscript is now acceptable for publication, you may indicate that here to bypass the “Comments to the Author” section, enter your conflict of interest statement in the “Confidential to Editor” section, and submit your "Accept" recommendation.

Reviewer #1: All comments have been addressed

2. Is the manuscript technically sound, and do the data support the conclusions?

Reviewer #1: Yes

3. Has the statistical analysis been performed appropriately and rigorously? 

Reviewer #1: Yes

4. Have the authors made all data underlying the findings in their manuscript fully available?

Reviewer #1: Yes

5. Is the manuscript presented in an intelligible fashion and written in standard English?

Reviewer #1: Yes

6. Review Comments to the Author

Reviewer #1: (No Response)

7. PLOS authors have the option to publish the peer review history of their article (what does this mean?). If published, this will include your full peer review and any attached files.

Reviewer #1: No

---

## [Editor Report · Acceptance letter]

17 Nov 2023

PONE-D-23-13221R3 

The effect of sodium-glucose cotransporter 2 inhibitors in patients with chronic kidney disease with or without type 2 diabetes mellitus on cardiovascular and renal outcomes: a systematic review and meta-analysis 

Dear Dr. Reategui-Diaz:

I'm pleased to inform you that your manuscript has been deemed suitable for publication in PLOS ONE. Congratulations! Your manuscript is now with our production department. 

Kind regards, 

on behalf of

Dr. Edward Zimbudzi 

Academic Editor

PLOS ONE